# Attitudes and Beliefs of Primary Care Physicians and Nurses in Spain Toward Vegan Diets

**DOI:** 10.3390/nu16233992

**Published:** 2024-11-21

**Authors:** Nuria Trujillo-Garrido, Eduardo Sánchez-Sánchez, María J. Santi-Cano

**Affiliations:** 1Faculty of Nursing, Nursing and Physiotherapy Department, University of Cádiz, 11207 Cádiz, Spain; eduardo.sanchez@uca.es; 2Institute of Biomedical Research and Innovation of Cádiz (INiBICA), 11009 Cádiz, Spain; mariajose.santi@uca.es; 3Faculty of Nursing and Physiotherapy, Nursing and Physiotherapy Department, University of Cádiz, 11003 Cádiz, Spain

**Keywords:** plant-based diets, healthcare professionals, perceptions, dietary counselling

## Abstract

Background: As dietary habits shift in response to environmental concerns and health awareness, understanding healthcare professionals’ perceptions of vegan diets is crucial. Objectives: This study aimed to identify the beliefs and attitudes of primary care doctors and nurses in Spain towards vegan diets. Methods: A questionnaire-based, observational, cross-sectional study was conducted among 208 healthcare professionals. Results: 87% of participants followed an omnivorous diet, while only 3.4% identified as lacto-ovo-vegetarian (LOV) or vegan. Statistically significant differences were observed by sex, with women more likely to agree that livestock farming contributes to global warming (27.3% and 28.0% vs. 17.0% and 12.8%, respectively; *p* = 0.02). Additionally, women were more inclined to consider vegan diets suitable for vulnerable groups, such as pregnant women and children, when adequately supplemented (24.8% and 17.4% vs. 10.6% and 10.6%, respectively; *p* = 0.030). Healthcare professionals who followed a LOV or vegan diet were more likely to disagree with the notion that vegan diets do not provide the necessary macronutrients and micronutrients compared to omnivores (strongly disagree 19.9%, 56.3%, 85.7%; *p* = 0.001 for omnivores, flexitarians, and LOVs/vegans, respectively). Conclusions: Current nutrition training may not meet the needs of doctors and nurses. Furthermore, it is implied that some professionals’ attitudes towards vegan diets may be more influenced by personal beliefs than by scientific literature. These findings can inform future clinical guidelines and support a more evidence-based approach to dietary counselling for vegan populations.

## 1. Introduction

It is now known that we are reaching the so-called planetary boundaries, understood as the global biophysical limits related to climate change, biosphere integrity, land-system change, and freshwater use [1]. Unsustainable food production in general, meat production in particular, is contributing to this, threatening the stability of the biosphere, with an estimated one-third of global greenhouse gas emissions coming from the food system [1,2]. The EAT-Lancet Commission on Healthy Diets from Sustainable Food Systems stated that “transformation to healthy diets by 2050 will require substantial dietary shifts, including a greater than 50% reduction in the global consumption of unhealthy foods such as red meat and sugar” [1]. This highlights the need to reflect on sustainable food production and healthy diets, leading the Food and Agriculture Organization (FAO) and the World Health Organization (WHO) to define what they mean by healthy and sustainable diets: “dietary patterns that promote all dimensions of individuals’ health and well-being; have low environmental impact; are accessible, affordable, safe, and equitable; and are culturally acceptable” [3].

Despite existing evidence regarding the impact of meat production on our environment, in developed countries, meat consumption is estimated to be three times higher than the recommended amount [4]. For example, 57% of men and 31% of women in the UK consume more than the recommended daily intake of red and processed meat [5]. However, in recent years it has also been observed that a growing number of omnivores are reducing their meat consumption [4,5,6,7,8,9,10]. While some European consumers are choosing to reduce their meat intake, not all evidence suggests that meat consumption is decreasing across European countries, as some regions continue to report stable or increasing levels [11,12]. In Europe, specifically, the sales value of plant-based foods increased by approximately 50% between 2018 and 2020, and more than half of all companies producing meat alternatives were founded in the last 10 years [13,14,15].

LOV and vegan diets, compared to omnivorous diets, can provide a significantly higher intake of health-promoting nutrients such as fiber, polyunsaturated fatty acids, alpha-linolenic acid, vitamins C, B1, B6, and E, folic acid, and magnesium [15]. In fact, Parker et al., in their systematic review, observed that LOVs and vegans featured a generally higher dietary quality on the Healthy Eating Index (HEI-2010) than those with omnivorous diets [16]. However, if any type of diet, including LOV and vegan diets, are not followed with specific nutritional knowledge, adequate planning, and thorough test-based follow-up, they can lead to micronutrient deficiencies in vitamin B12, omega-3 fatty acids, iron, iodine, zinc, and calcium [15,16,17].

Therefore, it is crucial that individuals willing to follow a LOV or vegan diet have access to appropriate dietary advice from healthcare professionals [9]. One of the most comprehensive tools available for this purpose is VegPlate, a Mediterranean-based food guide designed to assist health professionals in effectively planning well-balanced vegetarian and vegan diets across different life stages, including athletes, pregnant, and lactating individuals [18,19]. However, there are still few clinical practice guidelines developed by scientific societies providing doctors and nurses with the necessary support in terms of decision-making, which often makes these professionals feel uncertain or confused when a patient requests dietary advice [20,21,22,23].

Since primary care physicians and nurses are the first level of healthcare providers for the population and are key figures in disease prevention and promotion, it seems important to understand their beliefs and attitudes toward vegan diets, free of animal products. To our knowledge, there are no studies analyzing these issues in Spain, so the objective of our study was to identify the beliefs and attitudes of primary care doctors and nurses in Spain toward vegan diets.

## 2. Materials and Methods

### 2.1. Participants

This study involved a sample of primary care professionals, specifically doctors and nurses, actively practicing in Spain. The inclusion criteria were: primary care professionals (doctors and nurses) actively working in Spanish primary care centers and willing to provide informed consent. The exclusion criteria included professionals not currently practicing in primary care, as well as those who did not complete the entire questionnaire. A total of 208 participants took part in the survey.

### 2.2. Research Method

This observational, cross-sectional study utilized a questionnaire developed to assess healthcare professionals’ attitudes and beliefs regarding vegan diets. Data collection took place from March to June 2023. A cover letter was sent to all professional associations of physicians and nurses in Spain, as well as to scientific societies of primary care and community health. This letter included information such as the study’s name, the reason for such study, its objectives, its measurement tools, and its ethical considerations, inviting all professionals in the primary care field to participate. Additionally, the webpage link hosting the questionnaire (Google Forms) was included.

### 2.3. Data Collection Instrument

Since no validated scales were found after an exhaustive bibliographic search to meet the study objectives, a de novo questionnaire was drafted based on questionnaires from previously published studies [4,9,24,25]. The original questionnaire consisted of 21 questions, which were reduced to 10 after being tested by 10 experts in the field. Of these, 5 questions were redrafted to improve their understanding.

The questionnaire was divided into three sections. The first section aimed to collect demographic data. In the second section, questions enquired about participants’ assumptions regarding the relationship between diet and environment. The third section related to their beliefs and attitudes toward vegan diets. All the questions in the questionnaire may be found in Table 1. A 5-point Likert scale was used to evaluate the questionnaire: strongly agree; agree; neither agree nor disagree; disagree, and strongly disagree. A 5-option version was chosen, including the neither agree nor disagree response to avoid biased responses in any direction (Appendix A).

The study variables were: sex (female or male); age (categorized as follows: 20–30 years; 31–40 years; 41–50 years; 51–60 years; ≥61 years); profession (doctor or nurse); workplace location (urban: more than 50,000 inhabitants; semi-rural: between 5000 and 49,999 inhabitants; or rural: up to 4999 inhabitants); specific training in nutrition (formal university training: undergraduate or postgraduate degrees specifically in nutrition or dietetics; continuing education: ongoing professional development courses or training that healthcare professionals may complete outside of formal degree programs, continuing education training and no specific training); dietary habits: omnivorous (regular meat consumption); flexitarian (meat intake is reduced by occasionally abstaining from eating meat without completely giving up its consumption); and vegetarian/vegan (lacto-ovo vegetarians, vegetarians, and vegans), with vegans defined as those who exclude from their diet meat, eggs, dairy products, and any other animal-derived ingredients. This definition was provided to participants to ensure consistent understanding of the term [1,26,27].

### 2.4. Statistical Analysis

Statistical analysis was performed using SPSS 25. Since the variables were qualitative, non-parametric statistics were applied, using the Chi-Squared test to examine whether there were statistically significant differences between groups, with values of *p* < 0.05 considered significant. Building upon this, a logistic regression analysis and odds ratio calculations were performed, using each item of the questionnaire as dependent variables (agreement vs. disagreement), and age, sex, profession, location, education, and dietary habits as independent variables.

### 2.5. Ethical Considerations

This study was conducted in accordance with the Declaration of Helsinki (Fortaleza 2013). All participants signed an informed consent, without which they could not access the questionnaire. The consent form informed participants on the study objectives, the anonymous and voluntary nature of the questionnaire, and that they could opt for withdrawing at any time and request their data to be removed from the study. Participants were also required to read and agree to the Google Forms privacy policy before submitting their responses, ensuring awareness of data handling procedures. Additionally, they were also informed that collected data would only be used to analyze the study objectives and that none of the study authors were involved in any conflicts of interest.

## 3. Results

The sample consisted of 208 healthcare professionals, with 77.4% identifying as female. Among dietary habits, 87.0% reported following an omnivorous diet, 7.7% a flexitarian diet, and 3.4% a LOV or vegan diet. Age distribution varied by sex, with men showing a higher representation in the ≥61 age group (12.8%) compared to women (4.3%) (*p* = 0.032). Profession also differed significantly by sex, with 83.9% of women working as nurses while 31.9% of men were doctors (*p* = 0.017).

Regarding nutrition training, a higher percentage of men (68.1%) reported having no specific training in nutrition compared to women (55.1%). Within dietary groups, those following a LOV or vegan diet reported the highest proportion with continuing education in nutrition (57.1%), though differences in training across dietary habits were not statistically significant (Table 2).

Regarding the responses on the relationship between diet and environment, 27.3% and 28.0% of women agreed or strongly agreed that “livestock farming is one of the causes of global warming”, compared to 17.0% and 12.8% of men (*p* = 0.024). Additionally, 24.8% and 17.4% of women disagreed or strongly disagreed with the statement that “a vegan diet is unsuitable for pregnant women, breastfeeding mothers, children, or the elderly, even when supplemented with B12”, compared to 10.6% and 10.6% of men (*p* = 0.030). A total of 28.4%, 29.3%, 27.4%, and 30.8% of the entire sample reported to agree with statements related to the nutritional quality of vegan diets (“Following a vegan diet does not provide the necessary macronutrients and micronutrients”; “Following a vegan diet may lead to deficiencies in the recommended intake of protein”; “Following a vegan diet may lead to deficiencies in the recommended intake of iron”; and “Following a vegan diet may lead to deficiencies in the recommended intake of calcium”), with no statistically significant differences observed between both sexes (Table 3).

In the aggregated responses by gender, notable differences emerge across various questions. Women consistently showed a higher rate of agreement with the statements compared to men, particularly in Question 1, where 65.2% of women agree or strongly agree, versus 53.3% of men. Conversely, in questions like Question 5, a substantial proportion of both men and women express disagreement, highlighting a potential gender-based divergence in perception towards certain statements (Figure 1).

Regarding the participants’ responses according to their level of nutrition training, no statistically significant differences were found. A total of 52.8% of participants strongly disagreed with the statement, “If a healthy adult patient told me they follow a vegan diet, I would try to dissuade them from continuing”. Additionally, 35.6% and 30.7% of participants stated to strongly disagree and disagree, respectively, with the statement “I would be willing to replace natural meat with lab-grown meat when it becomes commercially available”.

The statements with which participants more generally agreed on were: “Livestock farming has a high environmental impact” (33.2%); “Following a vegan diet, free of animal products, with adequate nutritional knowledge and supplemented with vitamin B12, does not provide the necessary macronutrients and micronutrients” (28.3%); “Following a vegan diet, free of animal products, with adequate nutritional knowledge and supplemented with vitamin B12, may lead to deficiencies in the recommended intake of protein” (29.8%); “Following a vegan diet, free of animal products, with adequate nutritional knowledge and supplemented with vitamin B12, may lead to deficiencies in the recommended intake of iron” (27.8%); and “Following a vegan diet, free of animal products, with adequate nutritional knowledge and supplemented with vitamin B12, may lead to deficiencies in the recommended intake of calcium” (31.2%), with no statistically significant differences found between groups based on training (Table 4).

In the responses by training type, individuals with formal university training consistently show higher agreement levels across various statements compared to those with no specific training or continuing education. Notably, in Questions 1 and 2, the percentage of agreement for the university-trained group is markedly higher than in other groups, indicating a possible correlation between formal education and perception alignment (Figure 2).

Regarding the participants’ opinions based on their dietary habits, when responding to the statement “I would be willing to replace natural meat with lab-grown meat when it becomes commercially available”, 25% of flexitarians answered that they agree, and 28.6% of LOVs/vegans answered that they strongly agree, compared to 8.3% and 3.9% of omnivores, respectively (*p* = 0.028). Statistically significant differences were also observed between groups by dietary habits for the following statements: “Following a vegan diet is more expensive” (strongly disagree 8.8%, 25.0%, 57.1%; *p* = 0.001 for omnivores, flexitarians, and LOVs/vegans, respectively); “I believe that a vegan diet, free of animal products, is not suitable for pregnant women, breastfeeding mothers, children, or the elderly, even with adequate nutritional knowledge and supplemented with vitamin B12” (strongly disagree 12.7%, 31.3%, 71.4%; *p* = 0.003 for omnivores, flexitarians, and LOVs/vegans, respectively); “Following a vegan diet, free of animal products, with adequate nutritional knowledge and supplemented with vitamin B12, does not provide the necessary macronutrients and micronutrients” (strongly disagree 19.9%, 56.3%, 85.7%; *p* = 0.001 for omnivores, flexitarians, and LOVs/vegans, respectively); “Following a vegan diet, free of animal products, with adequate nutritional knowledge and supplemented with vitamin B12, does not provide the recommended intake of protein” (strongly disagree 14.9%, 56.3%, 71.4%; *p* = 0.001 for omnivores, flexitarians, and LOVs/vegans, respectively); “Following a vegan diet, free of animal products, with adequate nutritional knowledge and supplemented with vitamin B12, does not provide the recommended intake of iron” (strongly disagree 16.0%, 50.0%, 71.4%; *p* = 0.001 for omnivores, flexitarians, and LOVs/vegans, respectively); and “Following a vegan diet, free of animal products, with adequate nutritional knowledge and supplemented with vitamin B12, does not provide the recommended intake of calcium” (strongly disagree 18.2%, 56.3%, 71.4%; *p* = 0.003 for omnivores, flexitarians, and LOVs/vegans, respectively) (Table 5).

In the responses by diet type, vegetarians/vegans show notably higher agreement levels in Questions 1 and 2 compared to flexitarians and omnivores (Figure 3).

The logistic regression analysis did not identify models that adequately explained the responses to each item based on the studied variables. However, associations were observed for specific items. For Item 2, women were significantly more likely than men to consider livestock farming as one of the causes of global warming (odds ratio 3.87, *p* = 0.002). Regarding Item 3, flexitarians and vegans were less likely than habitual meat consumers to consider a vegan diet unsuitable for pregnant women, breastfeeding mothers, children, or the elderly (odds ratio 3.190, *p* = 0.033). For Item 4, flexitarians and vegans disagreed with the notion that a vegan diet is more expensive compared to habitual meat consumers (odds ratio 7.94, *p* = 0.000).

## 4. Discussion

The aim of this study was to explore the beliefs and attitudes of primary care physicians and nurses in Spain toward vegan diets, focusing on their perceptions of nutritional adequacy and sustainability.

Regarding the participant profile, the study sample was predominantly composed of female nurses working in urban areas, with most reporting an omnivorous diet, followed by a small proportion identifying as flexitarians or vegetarians/vegans. These figures are like those observed in the general population in Spain in 2023, where 9% of respondents reported to be flexitarian and 2.4% LOV and/or vegan [28].

Analyzing participants’ responses according to sex, statistically significant differences were found as women were more likely to agree with the statement “Livestock farming has a high environmental impact” and were more likely to disagree with the statement “I consider a vegan diet, free from animal products, unsuitable for pregnant women/breastfeeding mothers/children/elderly, even with adequate nutritional knowledge and supplemented with vitamin B12”. Moreover, women were significantly more likely than men to consider livestock farming as one of the causes of global warming (odds ratio 3.87, *p* = 0.002).

In alignment with other research findings, these gender-based differences reflect patterns in dietary attitudes observed internationally, where women generally show more favorable views toward plant-based diets [29,30,31]. For example, Knaapila et al. observed that women were more likely to restrict their consumption of animal products than men (42.5% vs. 18.8%), and women reported more positive associations with plant-based meat alternatives (and less positive associations with meat) than men [32]. Kemper et al. also found differences between sex and dietary habits, as both meat reducers (55%) and occasional meat consumers (63.4%) were more likely to be women, whereas meat consumers were more likely to be men [4]. Specifically in Spain, the study “The Green Revolution” by The Lantern consultancy also revealed changes in the LOV/vegan profile concerning sex, with women representing 74% of the LOV and vegan community [28].

In comparing beliefs based on nutrition training, no statistically significant differences were observed between groups. This may indicate that current professional training does not sufficiently influence healthcare professionals’ perceptions of plant-based diets, potentially leaving their attitudes largely unaffected by the extent of their nutrition education. A study by Metoudi et al. found that 79% of dietitians in the UK and Ireland felt they lacked sufficient training on plant-based diets and only 33% felt supported in recommending them at work [10]. Similarly, McHugh’s study on healthcare professionals’ beliefs about vegan diets showed that nearly half felt they had insufficient nutrition knowledge, with most advice given lacking clear, evidence-based information [33].

Differences in beliefs were observed based on dietary habits, particularly regarding the perceived nutritional adequacy of vegan diets. For example, most LOV/vegan participants strongly disagreed with the statement that a vegan diet does not provide necessary macronutrients and micronutrients when supplemented with vitamin B12, while only a small percentage of omnivores shared this view. A similar pattern was observed for concerns about deficiencies in protein, iron, and calcium.

These findings are consistent with those observed by other authors. In the aforementioned study by Metoudi et al. 48% of dietitians expressed concerns about the risk of malnutrition and micronutrient deficiencies. Additionally, 75% erroneously believed that plant proteins are incomplete and need to be combined [10].

Similarly, the findings of Villette et al., whose study aimed to describe the beliefs and attitudes of primary care physicians towards vegetarian diets, showed that the vast majority of physicians considered vegan diets to be associated with iron and protein deficiencies. Furthermore, participants in this study identified the risk of vitamin B12 deficiency in fourth place, only behind the risk of iron, calcium, and protein deficiencies, even though the only deficiency that is clearly described by scientific literature beyond any doubt and in all cases is vitamin B12 deficiency [16]. This could suggest that the interviewed physicians held false beliefs about the safety of such diets [9].

The widespread belief that plant-based proteins are inferior to animal proteins is notable. Protein quality depends on factors like digestibility, amino acid profile, food matrix, and processing [34]. However, some argue that methods for assessing protein quality may be biased toward animal sources, especially when total protein intake often exceeds recommendations [35]. Conversely, studies show that plant-based proteins can offer multiple benefits [14,36,37,38]. For instance, Carballo-Casla et al. found improved nutritional status among older adults when animal proteins were replaced with plant proteins [36]. Ardisson Korat et al. observed stronger associations between plant protein intake and the absence of physical limitations [37]. Similarly, Ortolá et al. concluded that higher intake of plant proteins may delay unhealthy aging [38]. Additionally, several studies suggest that reducing animal protein in favor of plant-based proteins promotes gut microbiota eubiosis, lowers inflammatory biomarkers, and decreases all-cause mortality [36,39,40].

Iron deficiency is a common concern in LOV and vegan diets. In the study by Kemper et al., over half of participants reducing meat intake continued some consumption, citing meat as a key iron source [41]. Although vegetarians often have lower iron stores, studies show no adverse health effects [9]; conversely, high heme iron intake is linked to greater coronary heart disease risk [33].

Calcium intake in vegan diets also remains a topic of debate in the scientific community. Some studies suggest a potential association between vegan diets and lower bone mass, possibly due to biases in the research [15]. Other research indicates no clear benefits from high dairy intake regarding bone strength, highlighting the need for further investigation [33].

In our results, statistically significant differences in attitudes toward vegan diets during vulnerable life stages emerged based on dietary habits. A high percentage of vegan and LOV participants disagreed with the statement that vegan diets are unsuitable for pregnant or breastfeeding women, children, or the elderly. In contrast, a much smaller percentage of omnivores disagreed with this statement with flexitarians and vegans showing greater likelihood of disagreeing compared to habitual meat consumers. These findings align with studies showing that vegan professionals are generally more supportive of plant-based diets during pregnancy, lactation, and childhood. These results align with those of Jeitler et al., who assessed healthcare professionals’ awareness of nutrient supplementation in vegan diets. They found that vegan professionals were more likely to support vegan nutrition during pregnancy, lactation, and childhood, with significant associations between dietary type and attitudes, possibly reflecting differences in interpretation of evidence [42]. Conversely, Villette et al. reported that most physicians viewed vegan diets as unsuitable for children, while Badasarre et al.’s review warned of possible deficiencies with vegan weaning if breastfeeding stops [9,27]. Other studies, however, support that well-planned vegan diets (with necessary supplements like vitamin B12) can provide balanced nutrition [43]. Overall, while evidence on vegan diets in vulnerable stages is inconclusive, there is consensus on the need for regular monitoring and supplementation, emphasizing the importance of training healthcare professionals to offer informed guidance [42,44].

Encouragingly, most participants stated they would not discourage patients from following a vegan diet, contrasting with findings from other studies where healthcare professionals showed more critical attitudes toward veganism [9,15,27,43]. In previous studies, significant portions of physicians reported advising against vegan diets, and parents of vegan children often faced dissuasion from healthcare providers. In this respect, Villette et al. found that half of surveyed physicians discouraged patients from adopting a vegan diet [9]. Similarly, Bivi et al. observed that many vegan parents received insufficient dietary advice for weaning, with 71% perceiving negative attitudes, such as criticism or dissuasion from doctors, and 36% choosing not to inform their pediatrician about vegan weaning [43]. Many parents even sought guidance from alternative sources due to conflicts with their doctors. Baldassarre et al.’s review echoed these findings, with 45% of parents reporting a lack of nutritional advice and 77% feeling discouraged from using alternative diets [27]. Borisova et al. also documented that many vegan adults experienced hostility from healthcare professionals, often leading them to conceal their dietary choices, which risks the health of vegan patients who may then seek unreliable sources or miss regular monitoring [15]. This communication gap may be due to vegans reporting feelings of criticism or pathologization from healthcare professionals, leading some to conceal their dietary choices. Indeed, the recent literature highlights a broader stigma toward patients with alternative diets, indicating social and professional challenges for vegans. According to Markowski et al., this stigmatization extends beyond the healthcare context, suggesting that vegans face similar biases in wider social settings [45].

Regarding attitudes toward lab-grown meat, flexitarian and vegan participants showed greater openness to considering lab-grown meat as a substitute for natural meat, aligning with other studies on attitudes toward alternative proteins [4]. However, research shows mixed results on this topic, pointing to the need for further studies to understand the variations in consumer attitudes fully [46,47]. There is a broad consensus that plant-based meat alternatives are the most accepted by consumers, as food neophobia is more often associated with options like insects or lab-grown meat [48,49,50,51]. These alternatives, often designed to appeal to flexitarians, may resemble meat but are typically ultra-processed and high in saturated fats and sodium, which limits their nutritional appeal [32].

In relation to cost perceptions of vegan diets, these also varied, with omnivores more likely to consider plant-based diets expensive, while flexitarians and vegans were less likely to perceive vegan diets as more expensive compared to habitual meat consumers. This belief may stem from the assumption that meat substitutes are necessary in vegan diets, overlooking affordable, nutrient-dense foods like legumes and nuts. In fact, the price of these marketed meat alternatives is one of the most cited barriers for those who want to adopt a vegan diet but do not [10,32,33,48].

### Limitations and Strengths

This study features some limitations. Firstly, the questionnaire might present selection and information biases within the target population. However, acquiescence—tendency to reply “yes”—was reduced by avoiding yes/no questions and the “don’t know” option as countermeasures for responses despite lack of knowledge, opinion, or relevance (non-attitudes). Additionally, while the questionnaire was developed following an exhaustive literature review on the subject, which provided key elements for the questions posed to participants, certain items may still carry an inherent response bias. This possible bias could subtly influence participants’ attitudes or beliefs, potentially affecting the objectivity of responses concerning beliefs and perceptions of vegan diets.

Secondly, our sample size was small compared to the population universe (primary care physicians and nurses in Spain); according to data from the Ministry of Health, in 2022, the Spanish National Health System (SNS) [52] had 36,239 doctors and 30,537 nurses affiliated with primary care teams. Our sample includes 41 doctors and 167 nurses, representing approximately 0.11% of primary care doctors and 0.55% of primary care nurses in Spain. Additionally, a limitation of our study is the predominance of nurses in the sample, which may lead to a stronger representation of nursing perspectives on vegan diets. In Spanish primary care, dietary counselling is shared between doctors and nurses, with doctors sometimes providing guidance directly or referring patients to nurses, depending on patient preferences and comfort levels. This emphasis on nursing views should be considered when interpreting the results, as doctors may approach dietary counselling differently within the primary care team. Furthermore, the small sample sizes of the flexitarian and vegan groups, particularly the vegan group with only seven participants, limit the statistical power of the comparisons between dietary groups, which may affect the generalizability of findings related to these specific groups. Thirdly, a potential limitation related to data security concerns arises from the survey’s online format. Although Google Forms was employed to distribute the questionnaire, which offered a convenient and accessible way to reach a broad participant base, it may present certain data privacy limitations. Google Forms does not always comply with stringent data protection standards, raising concerns regarding the confidentiality and security of responses. This limitation should be considered when interpreting the results, as it may impact participant privacy and data integrity. Finally, the professionals who participated might be those feeling most concerned about this topic, which could induce a recruitment bias. Therefore, this study presents an exploratory nature, providing information on the current situation and representing a starting point for future studies to delve into the attitudes and beliefs of healthcare professionals toward vegan diets.

## 5. Conclusions

Our findings indicate that female healthcare professionals demonstrate a higher awareness of the environmental impacts of diet and tend to hold a more favorable view of vegan diets’ health implications. The dietary habits of healthcare professionals appear to significantly shape their beliefs regarding vegan diets, sometimes more so than scientific evidence. This study also suggests that current nutrition training may not fully address the needs of healthcare professionals in primary care, as knowledge gaps and personal dietary beliefs influence their perceptions of vegan diets. Future research should focus on establishing clear clinical guidelines to support healthcare providers in delivering evidence-based dietary advice to patients who follow or are interested in plant-based diets.

## Figures and Tables

**Figure 1 nutrients-16-03992-f001:**
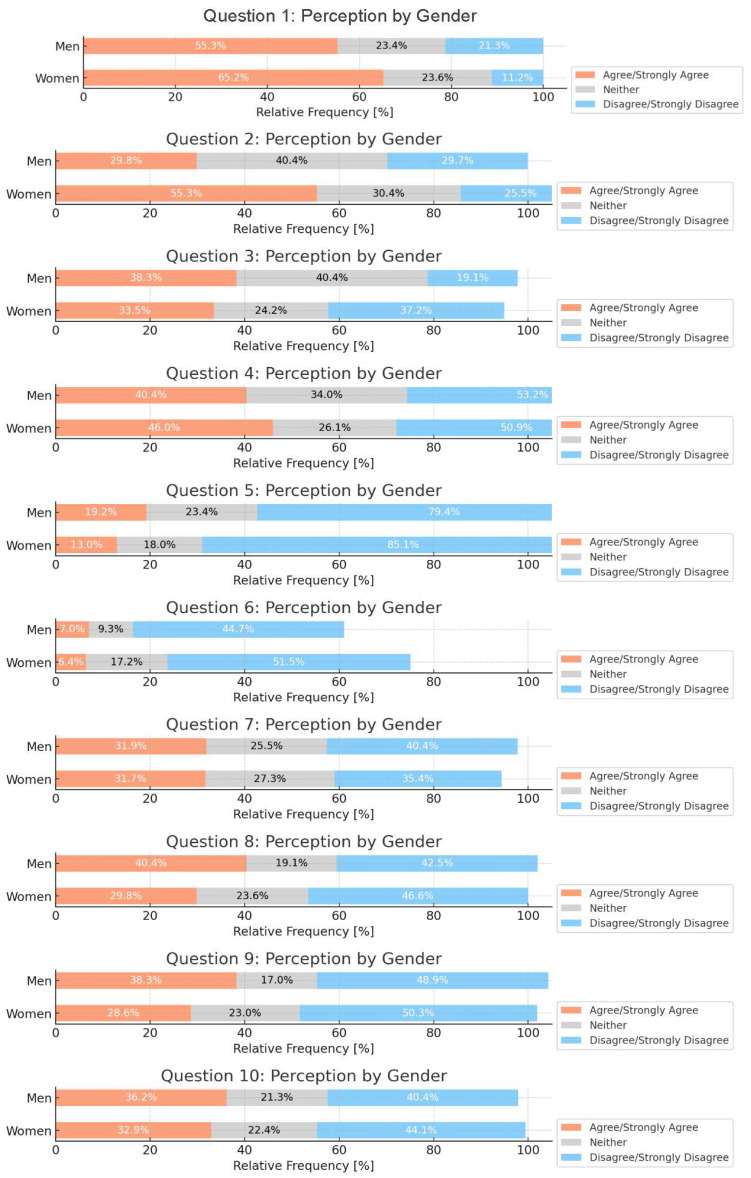
Aggregated responses by gender. Chi-squared test. Q1 *p* = 0.190; Q2 *p* = 0.005; Q3 *p* = 0.019; Q4 *p* = 0.562; Q5 *p* = 0.330; Q6 *p* = 0.448; Q7 *p* = 0.968; Q8 *p* = 0.387; Q9 *p* = 0.402; Q10 *p* = 0.918.

**Figure 2 nutrients-16-03992-f002:**
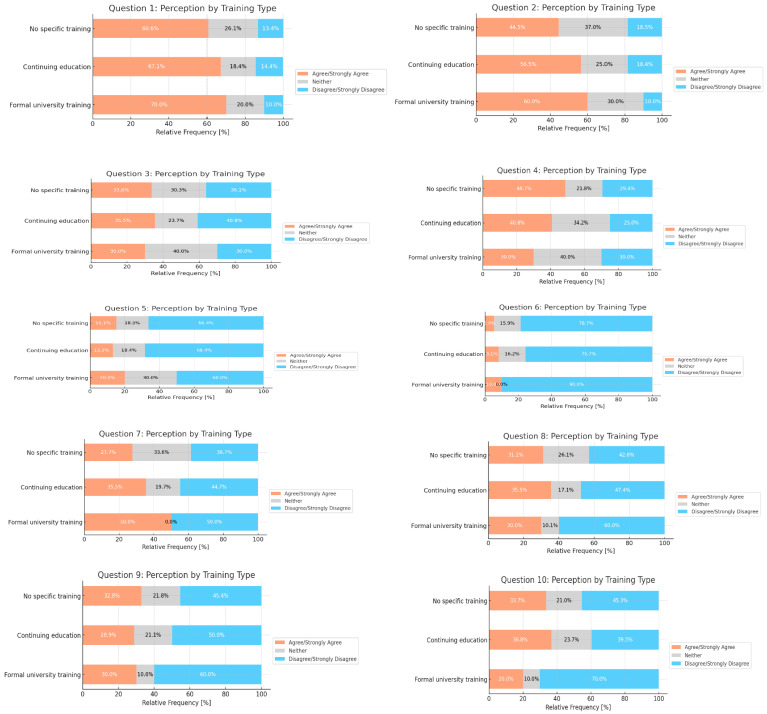
Aggregated responses by training. Chi-squared test. Q1 *p* = 0.782; Q2 *p* = 0.402; Q3 *p* = 0.782; Q4 *p* = 0.316; Q5 *p* = 0.836; Q6 *p* = 0.633; Q7 *p* = 0.068; Q8 *p* = 0.496; Q9 *p* = 0.849; Q10 *p* = 0.482.

**Figure 3 nutrients-16-03992-f003:**
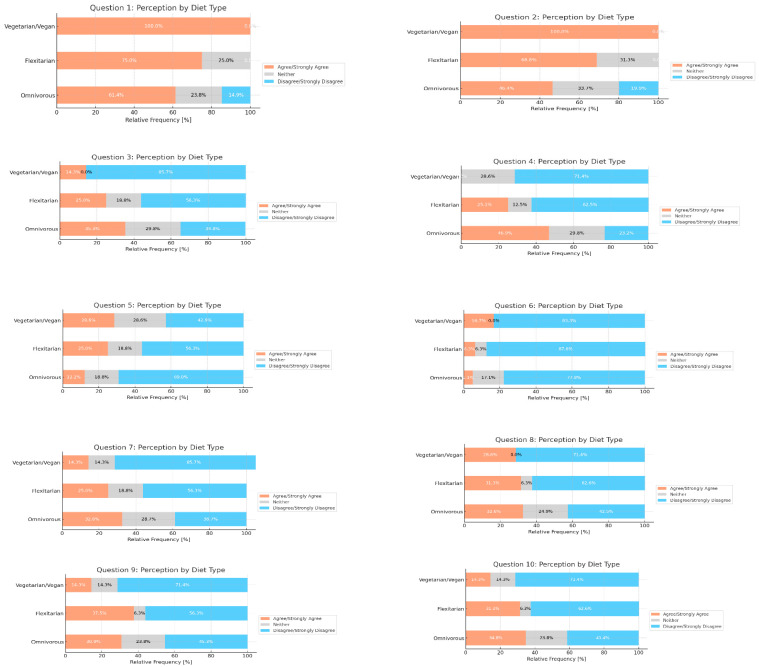
Aggregated responses by dietary habits. Chi-squared test. Q1 *p* = 0.134; Q2 *p* = 0.017; Q3 *p* = 0.040; Q4 *p* < 0.001; Q5 *p* = 0.371; Q6 *p* = 0.453; Q7 *p* = 0.095; Q8 *p* = 0.175; Q9 *p* = 0.357; Q10 *p* = 0.482.

**Table 1 nutrients-16-03992-t001:** Questionnaire items.

Question 1	Livestock farming has a high environmental impact.
Question 2	Livestock farming is one of the causes of global warming.
Question 3	I believe that a vegan diet (free from animal products) is not suitable for pregnant women, breastfeeding mothers, children, or the elderly, even when it is followed with adequate nutritional knowledge and supplemented with vitamin B12.
Question 4	Following a vegan diet (free from animal products) is more expensive.
Question 5	I would be willing to replace natural meat with lab-grown meat when it becomes commercially available.
Question 6	If a healthy adult patient told me they follow a vegan diet, I would try to dissuade them from continuing it.
Question 7	Following a vegan diet (free from animal products) with adequate nutritional knowledge and supplemented with vitamin B12 does not provide the necessary macronutrients and micronutrients.
Question 8	Following a vegan diet (free from animal products) with adequate nutritional knowledge and supplemented with vitamin B12 may lead to deficiencies in the recommended intake of protein.
Question 9	Following a vegan diet (free from animal products) with adequate nutritional knowledge and supplemented with vitamin B12 may lead to deficiencies in the recommended intake of iron.
Question 10	Following a vegan diet (free from animal products) with adequate nutritional knowledge and supplemented with vitamin B12 may lead to deficiencies in the recommended intake of calcium.

Author’s own work (2024).

**Table 2 nutrients-16-03992-t002:** General characteristics of the participants.

		Sex				Dietary Habits				
		Women (*n* = 161)	Men (*n* = 47)	Total (*n* = 208)	*p*	Omnivorous 87.0 (*n* = 181)	Flexitarian 7.7 (*n* = 16)	LOV/Vegan 3.4 (*n* = 7)	Total (*n* = 204) mv = 4	*p*
Age * % (*n*)	20–30	19.3 (31)	6.4 (3)	16.3 (34)	0.032	15.5 (28)	31.3 (5)	0.0 (0)	16.2 (33)	0.075
	31–40	24.2 (39)	14.9 (7)	22.1 (46)		21.5 (39)	12.5 (2)	71.4 (5)	22.5 (46)	
	41–50	27.3 (44)	36.2 (17)	29.3 (61)		29.3 (53)	31.3 (5)	14.3 (1)	28.9 (59)	
	51–60	24.8 (40)	29.8 (14)	26.0 (54)		26.5 (48)	25.0 (4)	14.3 (1)	26.0 (53)	
	≥61	4.3 (7)	12.8 (6)	6.3 (13)		7.2 (13)	0.0 (0)	0.0 (0)	6.4 (13)	
Profession % (*n*)	Doctor	16.1 (26)	31.9 (15)	19.7 (41)	0.017	19.9 (36)	25.0 (4)	0.0 (0)	19.6 (40)	0.366
	Nurse	83.9 (135)	68.1 (32)	80.3 (167)		80.1 (145)	75.0 (12)	100.0 (7)	80.4 (164)	
Workplace location % (*n*) mv = 2	Urban	62.3 (99)	78.7 (37)	66.0 (136)	0.112	66.5 (119)	43.8 (7)	85.7 (6)	65.3 (132)	0.294
	Semi-rural	26.4 (42)	14.9 (7)	23.8 (49)		23.5 (42)	37.5 (6)	14.3 (1)	24.3 (49)	
	Rural	11.3 (18)	6.4 (3)	10.2 (21)		10.1 (18)	18.8 (3)	0.0 (0)	10.4 (21)	
Specific training in nutrition % (*n*) mv = 3	Formal university training	5.1 (8)	4.3 (2)	4.9 (10)	0.277	5.6 (10)	0.0 (0)	0.0 (0)	5.0 (10)	0.664
	Continuing education	39.9 (63)	27.7 (13)	37.1 (76)		36.5 (65)	37.5 (6)	57.1 (4)	37.3 (75)	
	No specific training	55.1 (87)	68.1 (32)	58.0 (119)		57.9 (103)	62.5 (10)	42.9 (3)	57.7 (116)	

* Years; mv: missing value.

**Table 3 nutrients-16-03992-t003:** Questionnaire responses by sex.

		Woman (161)	Man (47)	Total (208)	*p*
Question 1	Strongly disagree % (*n*)	3.1 (5)	8.5 (4)	4.3 (9)	0.196
	Disagree % (*n*)	8.1 (13)	12.8 (6)	9.1 (19)	
	Neither agree nor disagree % (*n*)	23.6 (38)	23.4 (11)	23.6 (49)	
	Agree % (*n*)	31.7 (51)	36.2 (17)	32.7 (68)	
	Strongly agree % (*n*)	33.5 (54)	19.1 (9)	30.3 (63)	
Question 2	Strongly disagree % (*n*)	6.2 (10)	10.6 (5)	7.2 (15)	0.024
	Disagree % (*n*)	8.1 (13)	19.1 (9)	10.6 (22)	
	Neither agree nor disagree % (*n*)	30.4 (49)	40.4 (19)	32.7 (68)	
	Agree % (*n*)	27.3 (44)	17.0 (8)	25.0 (52)	
	Strongly agree % (*n*)	28.0 (45)	12.8 (6)	24.5 (51)	
Question 3	Strongly disagree % (*n*)	17.4 (28)	10.6 (5)	15.9 (33)	0.030
	Disagree % (*n*)	24.8 (40)	10.6 (5)	21.6 (45)	
	Neither agree nor disagree % (*n*)	24.2 (39)	40.4 (19)	27.9 (58)	
	Agree % (*n*)	26.7 (43)	23.4 (11)	26.0 (54)	
	Strongly agree % (*n*)	6.8 (11)	14.9 (7)	8.7 (18)	
Question 4	Strongly disagree % (*n*)	12.4 (20)	8.5 (4)	11.5 (24)	0.679
	Disagree % (*n*)	15.5 (25)	17.0 (8)	15.9 (33)	
	Neither agree nor disagree % (*n*)	26.1 (42)	34.0 (16)	27.9 (58)	
	Agree % (*n*)	29.2 (47)	21.3 (10)	27.4 (57)	
	Strongly agree % (*n*)	16.8 (27)	19.1 (9)	17.3 (36)	
Question 5	Strongly disagree % (*n*)	35.4 (57)	36.2 (17)	35.6 (74)	0.500
	Disagree % (*n*)	33.5 (54)	21.3 (10)	30.8 (64)	
	Neither agree nor disagree % (*n*)	18.0 (29)	23.4 (11)	19.2 (40)	
	Agree % (*n*)	9.3 (15)	12.8 (6)	10.1 (21)	
	Strongly agree % (*n*)	3.7 (6)	6.4 (3)	4.3 (9)	
Question 6	Strongly disagree % (*n*)	51.6 (81)	58.1 (25)	53.0 (106)	0.552
	Disagree % (*n*)	24.8 (39)	25.6 (11)	25.0 (50)	
	Neither agree nor disagree % (*n*)	17.2 (27)	9.3 (4)	15.5 (31)	
	Agree % (*n*)	4.5 (7)	2.3 (1)	4.0 (8)	
	Strongly agree % (*n*)	1.9 (3)	4.7 (2)	2.5 (5)	
Question 7	Strongly disagree % (*n*)	26.7 (43)	19.1 (9)	25.0 (52)	0.338
	Disagree % (*n*)	14.3 (23)	23.4 (11)	16.3 (34)	
	Neither agree nor disagree % (*n*)	27.3 (44)	25.5 (12)	26.9 (56)	
	Agree % (*n*)	29.2 (47)	25.5 (12)	28.4 (59)	
	Strongly agree % (*n*)	2.5 (4)	6.4 (3)	3.4 (7)	
Question 8	Strongly disagree % (*n*)	21.1 (34)	17.0 (8)	20.2 (42)	0.112
	Disagree % (*n*)	25.5 (41)	23.4 (11)	25.0 (52)	
	Neither agree nor disagree % (*n*)	23.6 (38)	19.1 (9)	22.6 (47)	
	Agree % (*n*)	28.6 (46)	31.9 (15)	29.3 (61)	
	Strongly agree % (*n*)	1.2 (2)	8.5 (4)	2.9 (6)	
Question 9	Strongly disagree % (*n*)	21.1 (34)	19.1 (9)	20.7 (43)	0.591
	Disagree % (*n*)	27.3 (44)	25.5 (12)	26.9 (56)	
	Neither agree nor disagree % (*n*)	23.0 (37)	17.0 (8)	21.6 (45)	
	Agree % (*n*)	26.1 (42)	31.9 (15)	27.4 (57)	
	Strongly agree % (*n*)	2.5 (4)	6.4 (3)	3.4 (7)	
Question 10	Strongly disagree % (*n*)	23.0 (37)	23.4 (11)	23.1 (48)	0.136
	Disagree % (*n*)	21.7 (35)	19.1 (9)	21.2 (44)	
	Neither agree nor disagree % (*n*)	22.4 (36)	21.3 (10)	22.1 (46)	
	Agree % (*n*)	31.7 (51)	27.7 (13)	30.8 (64)	
	Strongly agree % (*n*)	1.2 (2)	8.5 (4)	2.9 (6)	

Chi-squared test.

**Table 4 nutrients-16-03992-t004:** Questionnaire responses by nutrition training.

		Formal University Training	Continuing Education	No Specific Training	Total	*p*
		(10)	(76)	(119)	(205)	
Question 1	Strongly disagree % (*n*)	10 (1)	2.6 (2)	5.0 (6)	4.4 (9)	0.647
	Disagree % (*n*)	0.0 (0)	11.8 (9)	8.4 (10)	9.3 (19)	
	Neither agree nor disagree % (*n*)	20.0 (2)	18.4 (14)	26.1 (31)	22.9 (47)	
	Agree % (*n*)	30.0 (3)	31.6 (24)	34.5 (41)	33.2 (68)	
	Strongly agree % (*n*)	40.0 (4)	35.5 (27)	26.1 (31)	30.2 (62)	
Question 2	Strongly disagree % (*n*)	10 (1)	9.2 (7)	5.9 (7)	7.3 (15)	0.620
	Disagree % (*n*)	0.0 (0)	9.2 (7)	12.6 (15)	10.7 (22)	
	Neither agree nor disagree % (*n*)	30.0 (3)	25.0 (19)	37.0 (44)	32.2 (66)	
	Agree % (*n*)	30.0 (3)	27.6 (21)	23.5 (28)	25.4 (52)	
	Strongly agree % (*n*)	30.0 (3)	28.9 (22)	21.0 (25)	24.4 (50)	
Question 3	Strongly disagree % (*n*)	30.0 (3)	18.4 (14)	13.4 (16)	16.1 (33)	0.687
	Disagree % (*n*)	0.0 (0)	22.4 (17)	22.7 (27)	21.5 (44)	
	Neither agree nor disagree % (*n*)	40.0 (4)	23.7 (18)	30.3 (36)	28.3 (58)	
	Agree % (*n*)	20.0 (2)	26.3 (20)	25.2 (30)	25.4 (52)	
	Strongly agree % (*n*)	10.0 (1)	9.2 (7)	8.4 (10)	8.8 (18)	
Question 4	Strongly disagree % (*n*)	20.0 (2)	11.8 (9)	10.9 (13)	11.7 (24)	0.632
	Disagree % (*n*)	10.0 (1)	13.2 (10)	18.5 (22)	16.1 (33)	
	Neither agree nor disagree % (*n*)	40.0 (4)	34.2 (26)	21.8 (26)	27.3 (56)	
	Agree % (*n*)	20.0 (2)	25.0 (19)	29.4 (35)	27.3 (56)	
	Strongly agree % (*n*)	10.0 (1)	15.8 (12)	19.3 (23)	17.6 (36)	
Question 5	Strongly disagree % (*n*)	40.0 (4)	35.5 (27)	35.3 (42)	35.6 (73)	0.837
	Disagree % (*n*)	10.0 (1)	32.9 (25)	31.1 (37)	30.7 (63)	
	Neither agree nor disagree % (*n*)	30.0 (3)	18.4 (14)	18.5 (22)	19.0 (39)	
	Agree % (*n*)	20.0 (2)	7.9 (6)	10.9 (13)	10.2 (21)	
	Strongly agree % (*n*)	0.0 (0)	5.3 (4)	4.2 (5)	4.4 (9)	
Question 6	Strongly disagree % (*n*)	50.0 (5)	56.8 (42)	50.4 (57)	52.8 (104)	0.600
	Disagree % (*n*)	40.0 (4)	18.9 (4)	28.3 (32)	25.4 (50)	
	Neither agree nor disagree % (*n*)	0.0 (0)	16.2 (12)	15.9 (18)	15.2 (30)	
	Agree % (*n*)	10.0 (1)	5.4 (4)	2.7 (3)	4.1 (8)	
	Strongly agree % (*n*)	0.0 (0)	2.7 (2)	2.7 (3)	2.5 (5)	
Question 7	Strongly disagree % (*n*)	30.0 (3)	28.9 (22)	22.7 (27)	25.4 (52)	0.276
	Disagree % (*n*)	20.0 (2)	15.8 (12)	16.0 (19)	16.1 (33)	
	Neither agree nor disagree % (*n*)	0.0 (0)	19.7 (15)	33.6 (40)	26.8 (55)	
	Agree % (*n*)	40.0 (4)	31.6 (24)	25.2 (30)	28.3 (58)	
	Strongly agree % (*n*)	10.0 (1)	3.9 (3)	2.5 (3)	3.4 (7)	
Question 8	Strongly disagree % (*n*)	30.0 (3)	23.7 (18)	17.6 (21)	20.5 (42)	0.800
	Disagree % (*n*)	30.0 (3)	23.7 (18)	25.2 (30)	24.9 (51)	
	Neither agree nor disagree % (*n*)	10.1 (1)	17.1 (13)	26.1 (31)	22.0 (45)	
	Agree % (*n*)	30.0 (3)	31.6 (24)	28.6 (34)	29.8 (61)	
	Strongly agree % (*n*)	0.0 (0)	3.9 (3)	2.5 (3)	2.9 (6)	
Question 9	Strongly disagree % (*n*)	30.0 (3)	22.4 (17)	19.3 (23)	21.0 (43)	0.979
	Disagree % (*n*)	30.0 (3)	27.6 (21)	26.1 (31)	26.8 (55)	
	Neither agree nor disagree % (*n*)	10.0 (1)	21.1 (16)	21.8 (26)	21.0 (43)	
	Agree % (*n*)	30.0 (3)	25.0 (19)	29.4 (35)	27.8 (57)	
	Strongly agree % (*n*)	0.0 (0)	3.9 (3)	3.4 (4)	3.4 (7)	
Question 10	Strongly disagree % (*n*)	40.0 (4)	23.7 (18)	21.8 (26)	23.4 (48)	0.770
	Disagree % (*n*)	30.0 (3)	15.8 (12)	23.5 (28)	21.0 (43)	
	Neither agree nor disagree % (*n*)	10.0 (1)	23.7 (18)	21.0 (25)	21.5 (44)	
	Agree % (*n*)	20.0 (2)	34.2 (26)	30.3 (36)	31.2 (64)	
	Strongly agree % (*n*)	0.0 (0)	2.6 (2)	3.4 (4)	2.9 (6)	

Chi-squared test.

**Table 5 nutrients-16-03992-t005:** Questionnaire responses by dietary habits.

		Omnivorous (181)	Flexitarian (16)	Vegetarian/ Vegan (7)	Total (204)	*p*
Question 1	Strongly disagree % (*n*)	4.4 (8)	0.0 (0)	0.0 (0)	3.9 (8)	0.147
	Disagree % (*n*)	10.5 (19)	0.0 (0)	0.0 (0)	9.3 (19)	
	Neither agree nor disagree % (*n*)	23.8 (43)	25.0 (4)	0.0 (0)	23.0 (47)	
	Agree % (*n*)	34.3 (62)	25.0 (4)	28.6 (2)	33.3 (68)	
	Strongly agree % (*n*)	27.1 (49)	50.0 (8)	71.4 (5)	30.4 (62)	
Question 2	Strongly disagree % (*n*)	7.7 (14)	0.0 (0)	0.0 (0)	6.9 (14)	0.059
	Disagree % (*n*)	12.2 (22)	0.0 (0)	0.0 (0)	10.8 (22)	
	Neither agree nor disagree % (*n*)	33.7 (61)	31.3 (5)	0.0 (0)	32.4 (66)	
	Agree % (*n*)	24.3 (44)	37.5 (6)	28.6 (2)	25.5 (52)	
	Strongly agree % (*n*)	22.1 (44)	31.3 (5)	71.4 (5)	24.5 (50)	
Question 3	Strongly disagree % (*n*)	12.7 (23)	31.3 (5)	71.4 (5)	16.2 (33)	0.003
	Disagree % (*n*)	22.1 (40)	25.0 (4)	14.3 (1)	22.1 (45)	
	Neither agree nor disagree % (*n*)	29.8 (54)	18.8 (3)	0.0 (0)	27.9 (57)	
	Agree % (*n*)	26.5 (48)	25.0 (4)	0.0 (0)	25.5 (52)	
	Strongly agree % (*n*)	8.8 (16)	0.0 (0)	14.3 (1)	8.3 (17)	
Question 4	Strongly disagree % (*n*)	8.8 (16)	25.0 (4)	57.1 (24)	11.8 (24)	0.001
	Disagree % (*n*)	14.4 (26)	37.5 (6)	14.3 (1)	16.2 (33)	
	Neither agree nor disagree % (*n*)	29.8 (54)	12.5 (2)	28.6 (2)	28.4 (58)	
	Agree % (*n*)	28.7 (52)	18.8 (3)	0.0 (0)	27.0 (55)	
	Strongly agree % (*n*)	18.2 (33)	6.3 (1)	0.0 (0)	16.7 (34)	
Question 5	Strongly disagree % (*n*)	38.1 (69)	25.0 (4)	14.3 (1)	36.3 (74)	0.028
	Disagree % (*n*)	30.9 (56)	31.3 (5)	28.6 (2)	19.1 (39)	
	Neither agree nor disagree % (*n*)	18.8 (34)	18.8 (3)	28.6 (2)	19.1 (39)	
	Agree % (*n*)	8.3 (15)	25.0 (4)	0.0 (0)	9.3 (19)	
	Strongly agree % (*n*)	3.9 (7)	0.0 (0)	28.6 (2)	4.4 (9)	
Question 6	Strongly disagree % (*n*)	50.9 (89)	68.8 (11)	83.3 (5)	53.3 (105)	0.104
	Disagree % (*n*)	26.9 (47)	18.8 (3)	0.0 (0)	15.7 (31)	
	Neither agree nor disagree % (*n*)	17.1 (30)	6.3 (1)	0.0 (0)	15.7 (31)	
	Agree % (*n*)	3.4 (6)	6.3 (1)	0.0 (0)	3.6 (7)	
	Strongly agree % (*n*)	1.7 (3)	0.0 (0)	16.7 (1)	2.0 (4)	
Question 7	Strongly disagree % (*n*)	19.9 (36)	56.3 (9)	85.7 (6)	25.0 (51)	0.001
	Disagree % (*n*)	18.8 (34)	0.0 (0)	0.0 (0)	16.7 (34)	
	Neither agree nor disagree % (*n*)	28.7 (52)	18.8 (3)	14.3 (1)	28.4 (58)	
	Agree % (*n*)	29.3 (53)	25.0 (4)	14.3 (1)	28.4 (58)	
	Strongly agree % (*n*)	3.3 (6)	0.0 (0)	0.0 (0)	2.9 (6)	
Question 8	Strongly disagree % (*n*)	14.9 (27)	56.3 (9)	71.4 (5)	20.1 (41)	0.001
	Disagree % (*n*)	27.6 (50)	6.3 (1)	0.0 (0)	25.0 (51)	
	Neither agree nor disagree % (*n*)	24.9 (45)	6.3 (1)	0.0 (0)	22.5 (46)	
	Agree % (*n*)	29.8 (54)	31.3 (5)	28.6 (2)	29.9 (61)	
	Strongly agree % (*n*)	2.8 (5)	0.0 (0)	0.0 (0)	2.5 (5)	
Question 9	Strongly disagree % (*n*)	16.0 (29)	50.0 (8)	71.4 (5)	20.6 (42)	0.001
	Disagree % (*n*)	29.3 (53)	6.3 (1)	0.0 (0)	26.5 (54)	
	Neither agree nor disagree % (*n*)	23.8 (43)	6.3 (1)	14.3 (1)	22.1 (45)	
	Agree % (*n*)	27.6 (50)	37.5 (6)	14.3 (1)	27.9 (57)	
	Strongly agree % (*n*)	3.3 (6)	0.0 (0)	0.0 (0)	2.9 (6)	
Question 10	Strongly disagree % (*n*)	18.2 (33)	56.3 (9)	71.4 (5)	23.0 (47)	0.003
	Disagree % (*n*)	23.2 (42)	6.3 (1)	0.0 (0)	21.1 (43)	
	Neither agree nor disagree % (*n*)	23.8 (43)	6.3 (1)	14.3 (1)	22.1 (45)	
	Agree % (*n*)	32.0 (58)	31.3 (5)	14.3 (1)	31.4 (64)	
	Strongly agree % (*n*)	2.8 (5)	0.0 (0)	0.0 (0)	2.5 (5)	

Chi-squared test.

## Data Availability

The original contributions presented in the study are included in the article, further inquiries can be directed to the corresponding author.

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
