# Peer review of "Attitudes and Beliefs of Primary Care Physicians and Nurses in Spain Toward Vegan Diets"

_nutrients, 2024, doi:10.3390/nu16233992_

Round 1
Reviewer 1 Report
Comments and Suggestions for Authors
Comments and suggestions for Authors:
I appreciate the opportunity to review this article for Nutrients
Below, I provide a detailed analysis with general and specific comments that may enhance it:
ABSTRACT :
Line 12-13 - This text should be included in the Background/Objectives
INTRODUCTION
The introduction provided by the authors is quite comprehensive and addresses several relevant aspects of the study's topic.
General notes: Footnotes should be in square brackets. Avoid placing footnotes in the middle of sentences unless necessary. Please correct.
Line 51: While the authors are correct about limiting meat consumption among some EU populations, not all evidence suggests that meat consumption is decreasing in European countries.
https://www.statista.com/outlook/cmo/food/meat/europe, https://www.statista.com/topics/4197/meat-industry-in-europe/#topicOverview
MATERIALS AND METHODS :
First and foremost, the question arises whether the authors should consider forming a more homogeneous research group, specifically comprising either doctors or nurses. These are distinct professions with different qualifications, and the results obtained predominantly reflect the views of nurses, who participate in the study more frequently. It appears that doctors could have a greater influence on the nutritional education of patients.
There is some confusion in this subsection. It would be better to describe in more detail who took part in the research as study participants and in the next subsection present information about the research method.
Please describe the inclusion and exclusion criteria for the participants.
RESULTS :
The authors state that there are differences between the sexes, e.g. regarding the question: I consider a vegan diet -free from animal products- unsuitable for pregnant women/breastfeeding mothers/children/elderly, even with adequate nutritional knowledge and supplemented with vitamin B12. The group of men is much smaller and consists of 47 people. I wonder how many men from this group practice the profession of a nurse and how many are doctors. Were there men among the nurses? This needs to be clarified because perhaps groups of nurses and doctors should be considered separately in this study.
Table 4: There is no clear explanation of what continuing education means or what formal university education entails.
Table 5: The differences in the results of the answers to the questions between the groups are obvious. However, their statistical „strength” is low. The group of flexitarians, and especially vegans, is very small, consisting of only 7 people.
CONCLUSIONS:
The conclusion is overly lengthy and needs revision. The conclusions regarding nutrition training lack consistency.
Line 422-424 - This is not directly due to the results of the study. Please correct.
REFERENCES:
Please pay attention to the details and make corrections according to the journal's guidelines.

Author Response
Response to Reviewer 1 Comments
Dear Reviewer,
We would like to extend our sincere gratitude for the time and effort you have dedicated to reviewing our manuscript titled “Attitudes and Beliefs of Primary Care Physicians and Nurses Toward Vegan Diets.” Your insightful comments and suggestions have been invaluable in improving the quality and clarity of our work. We have carefully considered each of your recommendations and made the corresponding revisions. Below, we address your comments point by point, indicating the changes made to the manuscript.
Comment 1:
Abstract, Line 12-13 - This text should be included in the Background/Objectives.
Response 1:
Thank you for this observation. We agree that the information in line 12 provides essential context and fits better within the Background section of the abstract. We have moved this text accordingly to ensure a more structured and coherent presentation in line with your suggestion. Please see the changes highlighted in red in lines 13 and 14.
Comment 2:
Introduction
The introduction provided by the authors is quite comprehensive and addresses several relevant aspects of the study's topic.
General notes: Footnotes should be in square brackets. Avoid placing footnotes in the middle of sentences unless necessary. Please correct.
Response 2:
Thank you for this helpful suggestion. We have adjusted the citation format throughout the introduction, replacing parentheses with square brackets as recommended. Additionally, we ensured that citations are positioned at the end of sentences wherever possible to improve readability.
Comment 2 (part 2):
Line 51: While the authors are correct about limiting meat consumption among some EU populations, not all evidence suggests that meat consumption is decreasing in European countries.
https://www.statista.com/outlook/cmo/food/meat/europe, https://www.statista.com/topics/4197/meat-industry-in-europe/#topicOverview
Response 2.2:
Thank you for this observation. We recognize that trends in meat consumption across European countries can be complex, with some studies indicating reduction efforts while others show stable or even increased consumption in certain regions. To address this, we have revised the text to reflect a more nuanced perspective, acknowledging variability in meat consumption trends across Europe. We have also reviewed the resources you provided, added the references and adjusted our introduction accordingly. The changes can be found highlighted in red on lines 53 to 56.
Comment 4:
MATERIALS AND METHODS - Research Group Composition
First and foremost, the question arises whether the authors should consider forming a more homogeneous research group, specifically comprising either doctors or nurses. These are distinct professions with different qualifications, and the results obtained predominantly reflect the views of nurses, who participate in the study more frequently. It appears that doctors could have a greater influence on the nutritional education of patients.
Response 4:
Thank you for this insightful comment. We recognize the distinct qualifications and roles of doctors and nurses in patient care and the potential influence this may have on nutritional counselling. However, our study aimed to capture the attitudes and beliefs of primary care professionals collectively, reflecting the dynamics of both professions within the primary care setting. In Spain, dietary counselling is a recognized competence within nursing practice, and both doctors and nurses contribute actively to this aspect of patient care. Nutritional guidance is often a collaborative effort, with doctors sometimes opting to provide dietary advice directly, while in other cases preferring to refer patients to nursing professionals for in-depth counselling. The decision to refer may also be influenced by the specific relationships that patients have with different providers, as some patients may feel more comfortable discussing dietary habits with one professional over another.
In response to your comment regarding the composition of the research group, we conducted a statistical analysis (Chi-square test) to assess potential differences in responses between doctors and nurses. The results indicated no statistically significant differences between the two groups. A table summarizing this analysis has been added as supplementary material for reference.
Additionally, given the predominance of nurses in our sample, we have acknowledged this as a limitation in the discussion section to provide context for interpreting our results. Please refer to lines 436-442, which are highlighted in red.
Table Questionnaire responses by profession.
|
|
|
PROFESSION |
|||
|
|
|
DOCTOR (41) |
NURSE (167) |
TOTAL (208) |
p |
|
Q1 |
Strongly disagree % (n) |
4.9 (2) |
4.2 (7) |
4.3 (9) |
0.490 |
|
Disagree % (n) |
7.3 (3) |
9.6 (16) |
9.1 (19) |
||
|
Neither agree nor disagree. % (n) |
34.1 (14) |
21.0 (35) |
23.6 (49) |
||
|
Agree % (n) |
29.3 (12) |
33.5 (56) |
32.7 (68) |
||
|
Strongly agree % (n) |
24.4 (10) |
31.7 (53) |
30.3 (63) |
||
|
Q2 |
Strongly disagree % (n) |
7.3 (3) |
7.2 (12) |
7.2 (15) |
0.910 |
|
Disagree % (n) |
14.6 (6) |
9.6 (16) |
10.6 (22)
|
||
|
Neither agree nor disagree. % (n) |
31.7 (13) |
32.9 (55) |
32.7 (68) |
||
|
Agree % (n) |
22.0 (9) |
25.7 (43) |
25.0 (52) |
||
|
Strongly agree % (n) |
24.4 (10) |
24.6 (41) |
24.5 (51) |
||
|
Q3 |
Strongly disagree % (n) |
12.2 (5) |
16.8 (28) |
15.9 (33) |
0.512 |
|
Disagree % (n) |
14.6 (6) |
23.4 (39) |
21.6 (45) |
||
|
Neither agree nor disagree. % (n) |
31.7 (13) |
26.9 (45) |
27.9 (58) |
||
|
Agree % (n) |
34.1 (14) |
24.0 (40) |
26.0 (54) |
||
|
Strongly agree % (n) |
7.3 (13) |
9.0 (15) |
8.7 (18) |
||
|
Q4 |
Strongly disagree % (n) |
9.8 (4) |
12.0 (20) |
11.5 (24) |
0.484 |
|
Disagree % (n) |
7.3 (3) |
18.0 (30) |
15.9 (33) |
||
|
Neither agree nor disagree. % (n) |
31.7 (13) |
26.9 (45) |
27.9 (58) |
||
|
Agree % (n) |
29.3 (12) |
26.9 (45) |
27.4 (57) |
||
|
Strongly agree % (n) |
22.0 (9) |
16.2 (7) |
17.3 (36) |
||
|
Q5 |
Strongly disagree % (n) |
31.7 (13) |
36.5 (61) |
35,6% 74 |
0.288 |
|
Disagree % (n) |
36.6 (15) |
29.3 (49) |
30.8 64 |
||
|
Neither agree nor disagree. % (n) |
22.0 (9) |
18.6 (31) |
19.2 (40) |
||
|
Agree % (n) |
2.4 (1) |
12.0 (20) |
10.1 (21) |
||
|
Strongly agree % (n) |
7.3 (3) |
3.6 (6) |
4.3 (9) |
||
|
Q6 |
Strongly disagree % (n) |
50.0 (19) |
53.7 (87) |
53.0 (106) |
0.964 |
|
Disagree % (n) |
26.3 (10) |
24.7 (40) |
25.0 (50) |
||
|
Neither agree nor disagree. % (n) |
18.4 (7) |
14.8 (24) |
15.5 (31) |
||
|
Agree % (n) |
2.6 (1) |
4.3 (7) |
4.0 (8) |
||
|
Strongly agree % (n) |
2.6 (1) |
2.5 (4) |
2.5 (5) |
||
|
Q7 |
Strongly disagree % (n) |
24.4 (10) |
25.1 (42) |
25.0 (52) |
0.256 |
|
Disagree % (n) |
26.8 (11) |
13.8 823) |
16.3 (34) |
||
|
Neither agree nor disagree. % (n) |
24.4 (10) |
27.5 (46) |
26.9 (56) |
||
|
Agree % (n) |
19.5 (8) |
30.5 (51) |
28.4 (59) |
||
|
Strongly agree % (n) |
4.9 (2) |
3.0 (5) |
3.4 (7) |
||
|
Q8 |
Strongly disagree % (n) |
14.6 (6) |
21.6 (36) |
20.2 (42) |
0.704 |
|
Disagree % (n) |
29.3 (12) |
24.0 (40) |
25.0 (52) |
||
|
Neither agree nor disagree. % (n) |
19.5 (8) |
23.4 (39) |
22.6 (47) |
||
|
Agree % (n) |
31.7 (13) |
28.7 (48) |
29.3 (61) |
||
|
Strongly agree % (n) |
4.9 (2) |
2.4 (4) |
2.9 (6) |
||
|
Q9 |
Strongly disagree % (n) |
17.1 (7) |
21.6 (36) |
20.7 (43) |
0.947 |
|
Disagree % (n) |
26.8 (11) |
26.9 (45) |
26.9 (56) |
||
|
Neither agree nor disagree. % (n) |
22.0 (9) |
21.6 (36) |
21.6 (45) |
||
|
Agree % (n) |
29.3 (12) |
26.9 (45) |
27.4 (57) |
||
|
Strongly agree % (n) |
4.9 (2) |
3.0 (5) |
3.4 (7) |
||
|
Q10 |
Strongly disagree % (n) |
19.5 (8) |
24.0 (40) |
23.1 (48) |
0.806 |
|
Disagree % (n) |
24.4 (10) |
20.4 (34) |
21.2 (44) |
||
|
Neither agree nor disagree. % (n) |
24.4 (10) |
21.6 (36) |
22.1 (46) |
||
|
Agree % (n) |
26.8 (11) |
31.7 (53) |
30.8 (64) |
||
|
Strongly agree % (n) |
4.9 (2) |
2.4 (4) |
2.9 (6) |
||
Comment 5:
There is some confusion in this subsection. It would be better to describe in more detail who took part in the research as study participants and in the next subsection present information about the research method.
Response 5:
Thank you for this suggestion. We have restructured this section to clarify participant details and streamline the flow of information. Now, the subsection “Participants” details the study population, specifying that it involved 208 primary care professionals, including doctors and nurses practising in Spain, while the “Research Method” subsection focuses exclusively on describing the methodological approach. This restructuring enhances the clarity of the section and distinguishes between participant information and methodological details. Please see lines 84 to 91 highlighted in red.
Comment 6:
Please describe the inclusion and exclusion criteria for the participants.
Response 6:
We appreciate your attention to the inclusion and exclusion criteria. We have now specified these criteria in the “Participants” subsection. The inclusion criteria were: primary care professionals (doctors and nurses) actively working in Spanish primary care centres and willing to provide informed consent. The exclusion criteria included professionals not currently practising in primary care, as well as those who did not complete the entire questionnaire. These criteria are now explicitly stated to provide clarity on participant selection. Please refer to lines 86 to 90, which are highlighted in red.
Comment 7
RESULTS –
The authors state that there are differences between the sexes, e.g. regarding the question: "I consider a vegan diet -free from animal products- unsuitable for pregnant women/breastfeeding mothers/children/elderly, even with adequate nutritional knowledge and supplemented with vitamin B12." The group of men is much smaller and consists of 47 people. I wonder how many men from this group practice the profession of a nurse and how many are doctors. Were there men among the nurses? This needs to be clarified because perhaps groups of nurses and doctors should be considered separately in this study.
Response 7:
Thank you for this valuable observation. We agree that it is important to clarify the distribution of men and women across the professions in our sample. In our study, there were indeed male participants among both doctors and nurses, and we have now specified these details in the Results section to provide a clearer picture of the sample composition. Specifically, out of the 47 men in the study, 15 (31.9%) were doctors and 32 (68.1%) were nurses. Additionally, this information has been completed in Table 2 in the Results section.
Comment 8
Table 4: There is no clear explanation of what continuing education means or what formal university education entails.
Response 8:
Thank you for this observation. We appreciate the need for further clarity regarding the definitions of "continuing education" and "formal university education" in our study. In the current Methods section, these variables are already described; however, we will expand the information to make it even clearer. Specifically, continuing education refers to ongoing professional development courses or training that healthcare professionals may complete outside of formal degree programs. In contrast, formal university education includes undergraduate or postgraduate degrees specifically in nutrition or dietetics. Please refer to the lines 119 to 122 marked in red in the Methods section for these additions.
Comment 9
Table 5: The differences in the results of the answers to the questions between the groups are obvious. However, their statistical „strength” is low. The group of flexitarians, and especially vegans, is very small, consisting of only 7 people.
Response 9: Thank you for highlighting this issue. We recognize that the small sample size of flexitarian and vegan participants, especially the latter with only seven individuals, limits the statistical strength of the results and may affect the generalizability of our findings. Although we discuss limitations related to sample size and exploratory nature in the Discussion section, we have now expanded this to specifically acknowledge the impact of the small group sizes on statistical power. Please see lines 442-445 highlighted in red.
Comment 10
The conclusion is overly lengthy and needs revision. The conclusions regarding nutrition training lack consistency. Line 422-424 - This is not directly due to the results of the study. Please correct.
Response 10:
Thank you for your constructive feedback on the conclusions. We have revised the section to make it more concise and to improve the consistency of our statements regarding nutrition training. Additionally, we have addressed your specific comment regarding lines 422-424, and we have rephrased this part to ensure it more accurately reflects our intended meaning based on the results of our study. The revised conclusions are now included in the manuscript, highlighted in red for your convenience; please refer to the updated Conclusions section for review. The revised conclusions are as follows:
Conclusions: Our findings indicate that female healthcare professionals demonstrate a higher awareness of the environmental impacts of diet and tend to hold a more favourable view of vegan diets' health implications. The dietary habits of healthcare professionals appear to significantly shape their beliefs regarding vegan diets, sometimes more so than scientific evidence. This study also suggests that current nutrition training may not fully address the needs of healthcare professionals in primary care, as knowledge gaps and personal dietary beliefs influence their perceptions of vegan diets. Future research should focus on establishing clear clinical guidelines to support healthcare providers in delivering evidence-based dietary advice to patients who follow or are interested in plant-based diets.
Comment 11
REFERENCES: Please pay attention to the details and make corrections according to the journal's guidelines.
Response 11:
Thank you very much for your valuable feedback. We have implemented your suggestions by replacing all parentheses with square brackets for references throughout the document. Additionally, we have positioned references at the end of sentences to enhance readability.
We sincerely appreciate your valuable feedback. We trust that the revisions have adequately addressed your concerns and have improved the manuscript. Once again, thank you for your insightful comments and for your time dedicated to reviewing our work.
Sincerely,
Nuria Trujillo Garrido
Reviewer 2 Report
Comments and Suggestions for Authors
Dear Authors and Editors,
I am happy to help improving further the quality of this interesting and much needed manucript based on my subsequent comments.
Basically I feel this is a highly promising and important paper with a lot of potential (so far clearly not tapped full) - but only if its weak points will be fixed and diligently elaborated!
The formal presentation is nice, but Needs graphical presentation of results.
However, I am convinced after a very focused MAJOR revision of this manuscript it could be considered to stay in review process for publication.
All good success!
xxxxxxxxxxxxxxxxxxxx
Title:
ATTITUDES AND BELIEFS OF PRIMARY CARE PHYSI-2 CIANS AND NURSES TOWARD VEGAN DIETS (2 be added somehow: Spain or Spanish)
Keywords:
must not repeat keywords from the title, remove vegan diets; attitudes; beliefs; - find complemetary ones
xxxxxxxxxxxxxxxxxxxxxxxxxxxx
General comments:
Title must be extended by incluing Spain to e precise
the nice paper of Metoudi et al. (2024) needs to be added as reference,at least Intro and Disc:
A cross-sectional survey exploring knowledge, beliefs and barriers to whole food plant-based diets amongst registered dietitians in the United Kingdom and Ireland. https://doi.org/10.1111/jhn.13386
There is also other work from France, and from gyneologists - you might add (or not)
Formatting issue with Table 4 - pls fix
xxxxxxxxxxxxxxxxxxxx
Specific comments:
Intro:
line 60-63: this is true for any diet type - so pls add meaningful that misapplied or lack of Nutrition Knowledge Always results in Risk, this is not a Monopol of veggy or vegan diet per se - issue to be fixed/added just 1/2 sentence.
Method:
add the full questionnaire as Appendix or Supplement, Table 1 only gives a Rough Impression; however, formulation in Table 1 is not sober or neutral - issue that must be mentioned as Limitation as there might be affection/Impact on the Statements made by the participants
still unclear:
- was the survey online (if Google forms was used: this is a big data security problem and MUST be mentioned as marked mitation!) or paper-pencil provided - issue to be clarified in method section
- Definition of a vegan diet - as Item no. 7/Q 7 in Table 1 it is expected to the participant to know all the Details only from "free from animal products" that offers a wide Variation and range for personal Interpretation - issue to be fixed and Definition previously to be provided to Reader and clarified, if the paritipants got this specific Information and if so, in which way and extent
and: statistics seem very simple, even for a quantitive tool - why, why not a deeper dive for age, sex, Profession, DIET TYPE as independent or target or modulating variables etc.?
Results - Needs to be exact and sober, no Interpretation
I miss 1-2 nice figures - you have results that would be nice provided graphically and would break the text-table blocs in a nice way, adding to the Quality of the paper
line 126-127: remove "more thatn half of …." provide the EXACT nubmers, Always in Results section
Table 2 makes no sense at all - pls find guidance at other papers, eg. NURMI study papers on diet type - just to name 1 example of many: more colums/subroups: total - male - female - staticis and later: total mixed - vegetarian - vegan - statistics, just as 1 Approach that would help further - issue to be improved and fixed meaningfully!
also all Units are missing - one can just Image what is what …!
NEW and without Mention in the method = the diet type subgroups you present in Table 5 - there is NO reason given before and NO Definition for choosing a flexiatrian diet or pooling the veggies+vegans! therefore, providing the full survey as appendix or suplement is key! eyissue to be fixed and defintions to be provided!
Most tables could also be strenghtened by figures pooling the strongly disagree/disagree vs. strongly agree/agree so you have a clearer Picture, rather than presenting vast amounts of numbers, although informative - issue to be imnproved and beautified
Discussion - is rather Long and can be more to the Point (although solid argumentition), must be concise and must address all the main results - from here I cannot devide what are key results and Messages and what is additionally of interest and relevance - an interested Reader must be nicely guided by a clear structure - issue to be fixed for all discussion - issue of academic writing
Disc must be started with the aim of the study in reworded form, then followed by your main or key results 1-2-3-....6 - ? Idk - From then, proceed down all the key results - not just start with "direct attack" - issue to academic writing
lines 203-206 = Repetition to Introduction, leave it there in Intro and add defitions in intro
line 209: reword "Population under study" - sounds not very elegant
Limitations
- line 403: add the absolute number n and also the % dor total Spain doctors and also nurses to be able interprete 208 participants correctly considering Response rate (also here, add % of Response based on total Spain numbers)
- Needs to be Extended, see my comments Above
Conclusion is rather (too) Long - issue to compact to max. half page, better shorte
Reviewer 3 Report
Comments and Suggestions for Authors
This is a very interesting article exploring the attitudes and the beliefs towards vegan diets of health professionals in primary care in Spain. A cross-sectional survey has been performed by a 10-Q questionnaire proposed to 208 healthcare professionals. With the limit of any survey, although it resulted that women performed better on environmental and health knowledge, the inadequacy of the majority of the sample witnessed the need of a more science-based approach to this filed of nutrition, in order to provide valid counselling to vegan people.
General recommendations: please correct some typos
Introduction: please use lacto-ovo-vegetarian (in place of vegetarian) when it is intended, because of its ambiguity (sometimes the term “vegetarian” comprises lacto-ovo AND vegan). Lines 64-69: it’s worth mentioning here the VegPlate method, the most recent and complete food guide for health professionals allowing to well-planning vegetarian (i.e. lacto-ovo and vegan) diets for the main lifestages and for the athlete (PMID: 29170002; PMID: 30174286; PMID: 37049586).
Material and methods: 2.1. Study Design and Participants should include also 2.3. Data Collection
Results: the results are thoroughly presented, thank also to the very detailed tables
Discussion: line 419: a word is missing after “to” or the sentence is not clear. Please, rephrase it. This part is very long and repeats some of the results. I suggest proposing the concepts contained in the questions and discussing them.
Round 2
Reviewer 1 Report
Comments and Suggestions for Authors
Comments and suggestions for Authors:
I appreciate the opportunity to reread the article, "ATTITUDES AND BELIEFS OF PRIMARY CARE PHYSICIANS AND NURSES IN SPAIN TOWARD VEGAN DIETS” for consideration for publication in Nutrients.
The authors have addressed all my comments in their revisions and have clearly explained most of the doubts I previously had. While the study has notable limitations, the topic is interesting. The manuscript has significantly improved since the original submission. Therefore, I believe that the results are a valuable research contribution and may also encourage other researchers to further explore the issues discussed.
